The effect of gene polymorphism on ticagrelor metabolism: an in vitro study of 22 CYP3A4 variants in Chinese Han population

Hu Xiaoxia 15088913035@163.com 1
Wang Peng 2
Zeng Dali 3
Hu Guo-xin 4
1 Department of Pharmacy, Jinhua Municipal Central Hospital , Jinhua , China
2 Department of Pharmacy, Jinhua People’s Hospital , Jinhua , China
3 Department of Pharmacy, Wenzhou Hospital of Integrated Traditional Chinese and Western Medicine , Wenzhou , China
4 School of Pharmacy, Wenzhou Medical University , Wenzhou , China
Santillo Michael
Electronic publication date: 2024 Sep 24
Publication date: 2024
Volume: 12
Electronic Location ID: e18109
Received 2024 Jul 1; Accepted 2024 Aug 27
Copyright: ©2024 Hu et al.
Copyright year: 2024
Copyright holder: Hu et al.
License: This is an open access article distributed under the terms of the Creative Commons Attribution License, which permits unrestricted use, distribution, reproduction and adaptation in any medium and for any purpose provided that it is properly attributed. For attribution, the original author(s), title, publication source (PeerJ) and either DOI or URL of the article must be cited.
License URL: https://creativecommons.org/licenses/by/4.0/

Keywords: Ticagrelor, CYP3A4, Gene polymorphism, Drug metabolism, UHPLC-MS/MS

Funding: Jinhua Science and Technology Bureau of project No. 2023-3-121 This work was supported by the Jinhua Science and Technology Bureau of project (No. 2023-3-121). The funders had no role in study design, data collection and analysis, decision to publish, or preparation of the manuscript.

==============================
Background

Ticagrelor is a novel oral antiplatelet agent which can selectively inhibit P2Y12 receptor. Bleeding and dyspnea are common adverse reactions of ticagrelor in clinic. The side effects of ticagrelor are correlated with the plasma concentration of ticagrelor.

Objective

This study aimed to evaluate the catalytic characteristics of 22 CYP3A4 alleles identified in the Chinese Han population on the metabolism of ticagrelor in vitro, focusing on the effect of CYP3A4 polymorphism on ticagrelor metabolism.

Methods

In this study, insect cells were used to express 22 CYP3A4 variants, which were then incubated with 1–50 µM ticagrelor at 37 °C for 30 minutes to obtain the metabolite (AR-C124910XX). AR-C124910XX was detected by UHPLC-MS/MS to calculate the kinetic parameters, including Km, Vmax and CLint.

Results

Compared to the wild-type, most CYP3A4 alleles exhibited significant differences in intrinsic clearance. The intrinsic clearance of CYP3A4*11, *18 and *33 was much higher than that of wild-type; four variants exhibited similar intrinsic clearance values as the wild-type enzyme; The remaining 14 variants showed significantly reduced intrinsic clearance values, ranging from 1.48% to 75.11% of the wild-type; CYP3A4*30 displayed weak or no activity.

Conclusion

This study conducted a comprehensive assessment of the effect of CYP3A4 variants on ticagrelor’s metabolism. The results suggested that there is allele-specific activity towards ticagrelor in vitro. These findings can provide some insights and predictions for treatment strategies and risk assessments associated with ticagrelor in clinical practice.

Introduction

Ticagrelor, an oral antiplatelet agent of the cyclopentyltriazolopyrimidine class, is not a prodrug and therefore does not require metabolic activation (Khalil et al., 2024). It can directly, selectively and reversibly inhibit P2Y12 receptor through allosteric modulation, leading to a rapid and potent antiplatelet effect (Teng, 2015). Moreover, it reversibly blocks adenosine diphosphate (ADP) P2Y12 receptors, thus allowing the platelets activity to fully restore within three days after discontinuation of the therapy (Kabil, Abo Dena & El-Sherbiny, 2022). The instruction and previous researches indicate that ticagrelor can prevent atherothrombotic events in patients with acute coronary syndrome (ACS) or high-risk patients suffering from myocardial infarction, thereby reducing the incidence of cardiovascular death, myocardial infarction and stroke (Bonaca et al., 2015). The latest ESC guidelines recommended adding a P2Y12 receptor inhibitor to aspirin in all ACS patients; ticagrelor and prasugrel are recommended in preference to clopidogrel (Byrne et al., 2023).

While ticagrelor has shown remarkable efficacy in benefiting numerous patients with ACS, including those with clopidogrel resistance, there are still many reports of adverse reactions. One of the most significant adverse effects is varying degrees of bleeding. In a clinical trial, major bleeding was observed in 5.4% of cases in the ticagrelor group with 1989 patients (Schupke et al., 2019). CYP3A4 is the primary metabolic enzyme responsible for ticagrelor, contributing to 95% of its metabolism (Zhang et al., 2019). Ticagrelor undergoes dealkylation at the 5th position of the cyclopentane ring, forming the active metabolite AR-C124910XX, which exhibits systemic exposure levels approximately 30–40% that of ticagrelor (Al-Salama, Keating & Keam, 2017). Studies indicated that ticagrelor plasma concentrations, but not AR-C124910XX, were associated with bleeding events in Chinese patients with acute coronary syndrome (Yang et al., 2022). Wang et al. (2023) have pointed out that ticagrelor concentration exceeding 694.90 ng/mL was an independent risk factor for bleeding in ACS patients undergoing dual antiplatelet therapy, but the concentration of AR-C124910XX and salicylic acid were not associated with bleeding risk. Dyspnea is another common side effect of ticagrelor, leading to drug discontinuation in roughly 5% treated patients (Bonaca et al., 2015; Hiatt et al., 2017). A previous study supported that dyspnea is relevant to persistently higher plasma concentration of ticagrelor, but not that of AR-C124910XX (Ortega-Paz et al., 2018). Consequently, while both ticagrelor and its metabolites possess antiplatelet activity, the prevalent adverse reactions in clinical use are predominantly linked to ticagrelor plasma concentration. The possible reasons include that (1) the concentration of ticagrelor is much higher than that of AR-C124910XX; (2) the pharmacokinetic properties of ticagrelor differ from those of AR-C124910XX, including distribution and clearance (Liu et al., 2018). These findings underscore the importance of monitoring ticagrelor levels to anticipate risks of adverse reactions.

CYP3A4 is the responsible metabolizing enzyme of ticagrelor, and its gene polymorphisms could result in different plasma concentrations of ticagrelor in individuals at the same dose. CYP3A4 is highly polymorphic, exhibiting significant interindividual differences in enzyme activity, as evidenced by a 43.4-fold difference reported in previous research between the donors with the highest and lowest levels (Ohtsuki et al., 2012). So far, at least 54 CYP3A4 alleles (http://www.pharmvar.org/gene/CYP3A4) have been reported and named by the Human Cytochrome P450 Allele Nomenclature Committee. The frequencies of CYP3A4 variants exhibit considerable ethnic heterogeneity (Nicolas, Espie & Molimard, 2009). For instance, Sata et al. (2000) discovered that the frequency of the CYP3A4*1B allele was 4.2% in a white population, compared to 66.7% in a black population. This allele was found to be extremely rare among Chinese individuals, with the most common allele in the Han Chinese population being CYP3A4*1G, occurring at a frequency of 24.01%, which is deemed to wild-type variant in Chinese (Hu et al., 2017). Except for the wild-type, CYP3A4*18 is the most frequent allele in Chinese populations, whereas CYP3A4*22 predominates in Canadians (Holmberg et al., 2019; Hu et al., 2017). These genetic differences contribute to varying enzyme activities. Different metabolic enzyme activities are one of the most important reasons leading to treatment failure or toxic accumulation at normal doses (Hu et al., 2017). Several real-world studies have indicated that CYP3A4*22 carriers showed pronounced platelet inhibition after ticagrelor ingestion, and CYP3A4*22 variant was associated with increased risk of bleeding events in ticagrelor users (Holmberg et al., 2019; Liedes et al., 2023). Therefore, understanding the individual CYP3A4 gene polymorphisms is crucial for the safe use of ticagrelor. However, as far as we know, there is no comprehensively study about the effect of CYP3A4 polymorphism on ticagrelor metabolism.

In this study, we evaluated the effect of CYP3A4*1G and 22 CYP3A4 variants, including seven new variants that specifically distributed in Chinese Han population, on ticagrelor metabolism (Hu et al., 2017). CYP3A4 gene polymorphism is an important factor in drug metabolism. Given the clinical phenomenon of a significant correlation between ticagrelor concentration and adverse reactions, special attention should be paid to poor metabolizers (PMs). We hope that these results will help to identify PMs, expand the existing database on CYP3A4 gene polymorphisms, facilitate the development of metabolic models, and provide valuable reference for predicting in vivo ticagrelor concentration.

Materials & Methods

Chemicals and reagents

Ticagrelor (99% purity) was purchased from Shanghai Canspec Scientific & Technology Co., Ltd (Shanghai, China). Deshydroxyethoxy Ticagrelor (AR-C124910XX, 98% purity) was purchased from Toronto Research Chemicals Inc (Toronto, Ontario, Canada). The internal standard (IS) carbamazepine (98% purity) was obtained from Sigma-Aldrich Company (St. Louis, Mo, USA). Reduced NADPH was supplied by Roche (Basel, Switzerland). P450 cytochrome b5 and recombinant CYP3A4 microsomes expressed in baculovirus insect cell expression system was provided by Beijing Hospital (Beijing, China) (specific methods could be gotten in our previous research) (Liu et al., 2022). HPLC-grade acetonitrile and methanol were obtained from Merck (Darmstadt, Germany) and formic acid from J&K (Shanghai, China). The other chemicals were of analytical grade, and all solvents were of HPLC grade. Ultrapure water was prepared via a Milli-Q system (Millipore, Bedford, MA, USA).

Incubation conditions

Before incubation system established, enzyme concentration, incubation time and gradient concentration of ticagrelor were optimized. The protein concentration of recombinant CYP3A4 microsomes was determined using the carbon monoxide binding spectrum (Liu et al., 2022). The finial incubation system was confirmed, which included ticagrelor (1–50 µM), 5 µL CYP3A4 recombinant variants, 5 µL purified cytochrome b5, and PBS buffer (100 mM, PH 7.4). After 5 min pre-incubate in a Fisher shaking water bath, NADPH (1 mM) regenerating system was added to start the reaction at 37 °C water bath. The total volume was 200 µL. After 30 min, the incubation system was immediately terminated by cooling to −80 °C. Termination error of enzyme-catalyzed reaction was controlled by controlling the speed and duration of transshipment. We transferred samples to a −80 °C freezer for 15min to freeze them completely. Thus enzyme-catalyzed reaction could not restart in subsequent operations, which could ensure consistency of this study. After 15 min, 400 µL cool acetonitrile and 25 µL carbamazepine (400 µg/mL, IS) were added to the incubation mixture. Centrifuge the sample at 13,000 rpm for 10 min at 4 °C after fully vortexed. Then the supernatant 1:1 dilute with water, and 2 µL aliquot of the diluent was injected into an ultra-performance liquid chromatography tandem mass-spectrometer (UPLC-MS/MS) for analysis. All samples were prepared in triplicate parallelly.

Chromatography and UHPLC-MS/MS conditions

An Agilent 1290 system was used for the chromatographic separation of analytes on an Agilent ZORBAX RRHD Eclipse Plus C18 column (2.1 × 50 mm, 1.8 µ). The column temperature was maintained at 30 °C. Elution consisted of 0.l% (v/v) formic acid (A) and acetonitrile (B) were used as mobile phase to improve the peak shape and the separation efficiency. The flow rate was 0.4 mL/min. Gradient elution was optimized as follows: 0 min at 40% B, 0–0.5 min linear increased to 95% B, 0.5–1.5 min at 95% B, and 1.5–2.0 min linear decreased to 40% B. The total run time was 3.5 min with a stop-time of 2 min and a post-time of 1.5 min.

Detection and analysis were performed by an Agilent G6420A Triple quadrupole LC/MS equipped with electrospray ionization (ESI) source. AR-C124910XX and IS were tested in the multiple reaction monitoring (MRM) with positive mode. [M-H]+ was set as the precursor ion, and the most specific fragment ions were selected as the product ion (m/z 479.17 →153 for AR-C124910XX and m/z 237.1 →194 for IS). We set the capillary voltage at 4.0 kV, the nebulizer at 45 psi, the gas temperature at 350 °C, and nitrogen gas with a flow rate of 10L/min, respectively. Under these conditions, posttime retention times of AR-C124910XX and IS were 1.658 min and 0.859 min (Fig. 1), respectively. Data were acquired and quantified by Mass Hunter work station and Qualitative Analysis software (version B.07.00).

Figure 1 Representative chromatograms (IS) for carbamazepine and AR-C124910XX in a standard sample.

Posttime retention times of IS and AR-C124910XX were 0.859 min and 1.658 min.

Statistical analysis

The Michaelis–Menten curves and enzyme kinetic parameters of the 22 CYP3A4 recombinant enzymes were obtained using GraphPad Prism 5 (GraphPad Software Inc., San Diego, CA, USA) in the mode of nonlinear regression analysis. Enzyme kinetic parameters (Km and Vmax) for each variant were presented as the means ± SD from three parallel experiments. The intrinsic clearance (CLint) was determined as Vmax/Km. The Statistical Package for the Social Sciences (version 17.0; SPSS Inc., Chicago, IL, USA) was used to carry out statistical analysis. Prior to one-way ANOVA, the data were log-transformed to satisfy homogeneity of variance.; Levene’s test was used to evaluate homogeneity of variance; Statistically significant ANOVA results were followed by LSD post-hoc test. p < 0.05 represents statistically significant.

Results

The UHPLC-MS/MS method was verified with precision, accuracy, matrix effect, recovery and stability (Tables 1 & 2). The calibration curve established by linear regression model exhibited a good linear relationship over a concentration range of 5–2,000 nmol/L for AR-C 124910XX. The regression coefficients (R2) of calibration curve was above 0.99. The lower limit of quantification for AR-C124910XX was 5 nmol/L. Both the relative standard deviations (RSD) and relative errors (RE) of intraday and intreday precisions, and stability did not exceed ±15%. The recovery was 87.53%–91.29%. The matrix effect (ME) was 0.01%­-3.23%. The above results reflect that the analysis method is precise, accurate, stable, and which has a good recovery and a negligible matrix effect.

Table 1 Precision, accuracy, recovery and matrix effect for AR-C124910XX of QC sample in inactivated microsomal system (n = 6).

	Concentration added (nM)	Intra-day precision	Inter-day precision	Recovery (%)	Matrix effect (%)	
		RSD (%)	RE (%)	RSD (%)	RE (%)			
AR-C124910xx	20	13.54	8.42	2.65	8.39	87.53	3.23	
200	6.52	2.23	3.04	4.73	89.42	0.01	
2000	5.70	6.40	9.00	5.58	91.29	1.68	
Notes.

RSD, standard deviations; RE, relative errors

Table 2 Stability of AR-C124910XX in inactivated microsomal system under three different storage conditions (n = 6).

	Concentration added (nM)	Room temperature (4 h)	4 °C (24 h)	−80 °C (15 days)	
		RSD (%)	RE (%)	RSD (%)	RE (%)	RSD (%)	RE (%)	
AR-C124910xx	20	9.59	4.75	6.02	4.42	3.50	0.17	
200	6.94	2.29	8.66	−0.98	6.92	5.41	
2000	5.91	1.02	7.66	8.24	4.47	−2.47	
Notes.

RSD, standard deviations; RE, relative errors

By using this approach, we assessed the catalytic activities of wild-type CYP3A4 and 22 allelic variants with substrate ticagrelor. Michaelis–Menten plots for each CYP3A4 variant were shown in Fig. 2 and the corresponding kinetic parameters were summarized in Table 3. CYP3A4*30 (R130STOP) could not be determined because of no detectable enzymatic activity toward ticagrelor. Except for CYP3A4*30, the Michaelis–Menten kinetics of other 21 CYP3A4 variants could be determined. The estimated kinetic parameters Vmax, Km and CLint for AR-C124910XX of CYP3A4*1 were 1743.33 pmol/min/nmol, 2.18 µM and 800.45 µL/min/nmol, respectively. Almost of the variants exhibited changed Vmax or Km values compared with that of CYP3A4*1, which results in significant variations in CLint.

Figure 2 (A–E) Michaelis–Menten curve of the wild-type and 21 variants (R130STOP excepted) from ticagrelor metabolite (each point represents the mean ±S.D. of three parallel experiments).

Table 3 Kinetic parameters of ticagrelor metabolized by wild-type and 22 CYP3A4 variants.

Variants	cDNA changes	Main effect	Vmax (pmol/min/nmol P450)	Km (μM)	CLint (Vmax/Km) (μl/min/nmol P450)	Relative clearance (% of wild type)	
CYP3A4*1	/	/	1,743.33 ±50.29	2.18 ±0.17	800.45 ±40.05	100.00%	
CYP3A4*2	664T →C	S222P	5,703.33 ±284.59**	28.00 ±0.71**	203.59 ±5.45**	25.52%	
CYP3A4*3	1334T →C	M445T	2,082.33 ±147.55**	7.58 ±1.60**	280.23 ±39.47**	35.01%	
CYP3A4*4	352A →G	I118V	3,587.33 ±496.24**	6.55 ±1.28**	551.89 ±31.42**	68.84%	
CYP3A4*5	653C →G	P218R	7,279.00 ±53.69**	19.70 ±0.82**	369.89 ±16.65**	46.25%	
CYP3A4*9	508G →A	V170I	2,882.00 ±173.50**	4.80 ±0.25**	599.66 ±5.35**	74.97%	
CYP3A4*10	520G →C	D174H	5,238.33 ±355.92**	5.90 ±0.52**	889.66 ±18.35	111.42%	
CYP3A4*11	1088C →T	T363M	6,714.33 ±370.00**	3.18 ±0.36**	2,123.33 ±121.24**	264.73%	
CYP3A4*14	44T →C	L15P	4,655.67 ±309.74**	20.30 ±2.03**	229.84 ±7.89**	28.74%	
CYP3A4*15	485G →A	R162Q	3,188.67 ±33.01**	5.58 ±0.34**	573.55 ±40.52**	71.80%	
CYP3A4*16	554C →G	T185S	2,801.33 ±283.31**	7.79 ±0.97**	360.07 ±8.32**	45.07%	
CYP3A4*17	566T →C	F189S	382.67 ±23.28**	10.17 ±1.01**	37.75 ±1.70**	4.72%	
CYP3A4*18	878T →C	L293P	3,168.00 ±118.51**	3.24 ±0.05**	977.79 ±28.78**	122.49%	
CYP3A4*19	1399C →T	P467S	2,437.67 ±20.55**	4.56 ±0.55**	539.74 ±65.26**	67.37%	
CYP3A4*23	484C →T	R162W	3,990.33 ±164.24**	5.42 ±0.76**	742.86 ±73.74	92.82%	
CYP3A4*24	600A →T	Q200H	42.54 ±2.75**	3.57 ±0.32**	11.92 ±0.31**	1.48%	
CYP3A4*28	64C →G	L22V	1,580.00 ±64.37	3.76 ±0.56**	425.14 ±46.65**	53.32%	
CYP3A4*29	337T →A	F113I	1,697.33 ±25.79	3.20 ±0.06**	531.30 ±5.91**	66.46%	
CYP3A4*30	388C →T	R130STOP	N.D.	N.D.	N.D.	N.D.	
CYP3A4*31	972C →A	H324Q	2,504.33 ±106.33**	4.17 ±0.27**	601.16 ±13.90**	75.11%	
CYP3A4*32	1004T →C	I335T	3,329.33 ±304.63**	4.06 ±0.86**	833.05 ±97.79	103.77%	
CYP3A4*33	1108G →T	A370S	4,590.67 ±342.15**	4.88 ±0.36**	940.29 ±7.51**	117.77%	
CYP3A4*34	1279A →G	I427V	3,599.33 ±59.52**	4.04 ±0.18**	893.15 ±53.76	111.93%	
Notes.

* p < 0.05 versus wild-type.

** p < 0.01 versus wild-type.

ND, not determined.

According to the alterations in Vmax value, three situations were observed: significant decrements in CYP3A4*17 and CYP3A4*24 compared with CYP3A4*1; no obvious differences in CYP3A4*28 and CYP3A4*29; significant increments in the other 17 variants. Km values of all variants showed statistically increased (1.46–12.84 times of the wild-type). The ratio of Vmax and Km indicates the intrinsic clearance (CLint) values, which are considered the evaluation criterion for enzymatic activity. The relative clearance values between variants and wild-type can be determined by comparing their CLint ratios. Except for CYP3A4*30, the remaining 21 CYP3A4 variants could be classified into the following three categories according to the relative clearance values compare with CYP3A4*1: CYP3A4*11, CYP3A4*18 and CYP3A4*33 showed significantly higher intrinsic clearance than that of CYP3A4*1; four CYP3A4 variants (CYP3A4*10, CYP3A4*23, CYP3A4*32 and CYP3A4*34) were not significantly different from that of wild-type; the remaining fourteen variants (CYP3A4*2-*5, *9, *14-*17, *19, *24, *28, *29, and *31,) exhibited significantly reduced CLint in different degrees (1.48%–75.11% relative clearance) compared to that of wild-type.

Discussion

CYP3A4 is considered the most predominant CYP450 enzyme in the human liver and the gut. It is responsible for the metabolism of approximately 30–40% of clinically used drugs (Hu et al., 2017). CYP3A4 exhibits high polymorphism, with different enzyme activities closely related to factors such as race, genetic variation, and interactions with substrates, inhibitors, or inducers. The enzyme catalytic activity of CYP3A4 alleles can be categorized into four categories: no activity, decreased activity, normal activity and increased activity. Individuals carrying inactive or decreased active CYP3A4 alleles are referred to as poor metabolizers (PMs); while those who carry increased active CYP3A4 alleles are referred to as extensive metabolizers (EMs). Compared to normal metabolizes, PMs are more susceptible to medication side effects at normal therapeutic doses, particularly for drugs with narrow therapeutic windows or large individual variability; conversely, EMs are more likely to experience treatment failure at normal doses.

Research conducted over the past decade has revealed that while there are no significant differences in the antiplatelet efficacy of ticagrelor across different genotypes, there is substantial variation in the risk of adverse effects such as bleeding complications and dyspnea (Ortega-Paz et al., 2018; Schupke et al., 2019; Wang et al., 2023). This variation arises because ticagrelor and its main metabolite, AR-C124910XX, have comparable antiplatelet activity. However, the adverse effects following clinical administration of ticagrelor are correlated with the plasma concentration of ticagrelor, but not of AR-C124910XX (Ortega-Paz et al., 2018; Schupke et al., 2019; Wang et al., 2023; Yang et al., 2022). Accurate individualized drug dosing is crucial for reducing adverse events. CYP3A4 gene polymorphism is a key factor that leads to different substrate concentrations (Baxter et al., 2014). At equivalent dosages, individuals classified as poor metabolizers (PMs) exhibit higher concentrations of ticagrelor, thereby increasing their risk of bleeding and dyspnea. In contrast, extensive metabolizers (EMs) or those with normal metabolic function have a lower likelihood of experiencing these adverse effects.

CYP3A4*17 (F189S) and *24 (Q200H) exhibit severely reduced activity in the metabolism of ticagrelor, with over 90% decreased CLint compared to the wild-type. Similar reductions in metabolic activity have been observed with other substrates, such as sildenafil and ibrutinib (Tang et al., 2020; Xu et al., 2018). CYP3A4*17 (F189S) is a typical activity-attenuated variant. Dai et al. (2001) predict that residue F189S is located at the end of helix E. The residue is a nonconservative mutation in a tightly packed region, which could potentially impact the conformation of the protein, substrate access, and/or catalytic activity (Dai et al., 2001). Our results provide some support for this prediction, indicating that CYP3A4*17 exhibits a markedly reduced Clint for ticagrelor, achieving only 4.72% of the activity observed in the wild-type. CYP3A4*24 (Q200H) involves an A →T nucleotide substitution in exon 7, which causes a change from glutamine to histidine at position 220. In this study, CYP3A4*24 showed feeble catalytic activity towards ticagrelor, with only 1.48% relative clearance. The precarious activity of CYP3A4*24 may be attributed to the Q200H mutation.

Twelve variants (CYP3A4*2, *14, *3, *16, *5, *28, *29, *19, *4, *15, *9 and *31) also display significantly lower catalytic activity than CYP3A4*1, with relative clearance ranging from 25.52% to 75.11%. CYP3A4*2 is initially identified in a white population with a frequency of 2.7% (Sata et al., 2000), and has since been observed in the Chinese population. It involves a 15713T>C transition resulting in a Ser222Pro change. The serine to proline amino acid change could potentially affect the three-dimensional structure of the protein due to proline’s known ability to disrupt helical structures (Lee & Goldstein, 2005). Therefore, the mutation of Ser222Pro may greatly affect the substrate binding process and result in a relatively reduced catalytic activity of CYP3A4*2. In this in vitro study, CYP3A4*2 exhibits a 74.48% decrease in intrinsic clearance for ticagrelor when compared to the wild-type. CYP3A4*3 was first identified in Chinese subject from Shanghai (Sata et al., 2000), and it has a prevalence of 0.5% in the Uygur population in northwest China (Jin et al., 2015). It involves a 1334T>C transition leading to a missense mutation with a Met445Thr substitution in exon 12. This SNP take place within an evolutionarily conserved heme-binding region (Sata et al., 2000), potentially inducing structural variations affecting enzymatic activity. This study confirmed this point: the increase in Km is larger than that of Vmax, resulting in a 64.99% decrease in CLint for CYP3A4*3 compared with CYP3A4*1. Patients carrying these activity-impaired alleles should be classified as poor metabolizers. Previous studies have reported similar variants with reduced catalytic activity. For example, the CYP3A4*22 allele defined by the rs35599367 variant, which reduces CYP3A4 mRNA expression and enzyme activity in the liver (Van Eerden et al., 2023). Research reported an 89% higher area under the plasma concentration–time curve for ticagrelor in CYP3A4*22 carriers compared to controls, suggesting that the CYP3A4*22 carriers may reinforce the antiplatelet effect by impairing ticagrelor elimination (Holmberg et al., 2019). Thus, we speculate that carriers of the aforementioned activity-impaired variants, similar to CYP3A4*22 carriers, may exhibit higher concentrations of ticagrelor than wild-type carriers, leading to more substantial effects. Therefore, they should be more vigilant regarding adverse reactions such as bleeding and dyspnea. Patients who carry these activity-impaired alleles may require lower ticagrelor doses to achieve therapeutic plasma concentrations and reduce the risk of adverse effects and toxicity.

CYP3A4*11, *18, and *33 show significantly higher catalytic activities compared to the wild-type. CYP3A4*11 contains a substitution from threonine to methionine at residue 363 site. This substitution might perturb the tertiary structure of the protein in the region, resulting in instability of its catalytic function (Eiselt et al., 2001). In contrast to the current findings, Murayama et al. (2002) indicated that CYP3A4*11 exhibited reduced catalytic activity in testosterone hydroxylase activity compared with CYP3A4*1 (Eiselt et al., 2001). CYP3A4*18 is the most common nonsynonymous variant in the Han Chinese population with a frequency of 1.26% (Hu et al., 2017). CYP3A4*18 contains one nucleotide substitution (T>C) at position 878 in the cDNA, which results in an amino acid change from Leucine to Proline at position 293. Codon 293 is located at the start of the highly conserved helix I, in which the most crucial difference in the L293P secondary structural elements was observed (Kang et al., 2009). Previous in vitro functional studies indicate that CYP3A4*18 results in a rapid oxidation in sex steroid metabolism (Kang et al., 2009). In current study, CYP3A4*11 and CYP3A4*18 exhibit apparent increased intrinsic clearance values for ticagrelor. These different results may be explained by that the structural changes in the substrate recognition sites of the CYP450 enzyme that can cause changes in enzymatic activity.

CYP3A4*30 is a novel allelic variant found in the Chinese population. In experiments involving ticagrelor incubation, the enzymatic activity of CYP3A4*30 could not be assessed. Studies have revealed that the nucleotide changes of c.388C>T (CYP3A4*30) leads to a premature termination codon at position 130, resulting in a truncated protein that lacks the catalytic domain (Hu et al., 2017). Thus, CYP3A4*30 may exhibit complete loss of function.

Almost all CYP3A4 alleles exhibit altered or decreased functions compared to wild-type. It should be noticed that one particular variant can display decreased, increased, or unchanged activities for different drugs. Namely, the substrate specificity is an important reason leading to altered characteristic of the CYP3A4 variants. Thus, it is necessary to carry out enzyme kinetics studies on different substrates.

It’s noteworthy that the clinical Cmax for ticagrelor is approximately 1 µM, which is significantly lower than the Km values of each CYP3A4 variant (Wang et al., 2021). This suggests that the in vivo metabolic rate of ticagrelor is far from Vmax and likely follows first-order kinetics typical of enzyme-catalyzed reactions, indicating ticagrelor is within its linear metabolic range. Poor metabolizers (PMs) of CYP3A4 exhibit reduced metabolism, resulting in prolonged drug half-life and increased plasma concentrations of ticagrelor, potentially leading to higher incidence of adverse events. For the Chinese population carrying these variants, particularly PMs, understanding enzyme kinetic characteristics (such as Clint) can be useful in predicting ticagrelor toxicity and maintaining vigilance in clinical practice. Future clinical studies focusing on the pharmacokinetics of ticagrelor in typical poor metabolizers are needed. Our goal in future research is to establish a drug concentration calculation model based on enzyme kinetics and pharmacokinetic characteristics to advance precision medicine.

Conclusions

In summary, this study provides a functional assessment of 22 recombinant CYP3A4 variants in ticagrelor metabolism. Variants in CYP3A4 exhibit substrate-specific differences, for which it is necessary to study the effect of various variants towards different substrates. Most of the variants showed markedly different enzyme catalytic activity in ticagrelor metabolism in vitro. The side effects in clinical of ticagrelor are correlated with the plasma concentration of ticagrelor. It demonstrates that monitoring serum concentration and adjusting ticagrelor dosage in time is vital for the patients carrying these variants to achieve sound therapeutic effects avoiding side effects. We have reason to speculate that patient’s CYP3A4 genotype may exert a meaningful impact on the therapeutic effect and side effect of ticagrelor. We hope that these results could extend the understanding of the clinical drug toxicity, and provide insights and predictions for treatment strategies and risk assessments associated with ticagrelor in clinical practice to some extent.

Supplemental Information

Supplemental Information 1 Parameters of ticagrelor metabolized by wild-type and 22 CYP3A4 variants

Yellow highlight is the concentration of the variant. The first column shows the variant at different concentrations. Column 2-4 show the AR-C 1249100xx concentrations determined by UHPLC-MS/MS. Column 5-7 show the calculated metabolic rates. Column 8 and Column 9 are mean and SD value of Column 5-7.

Supplemental Information 2 Original images of Figure 1

The authors thank the members of the Beijing Institute of Geriatrics of the Ministry of Health for their advice and assistance.

Additional Information and Declarations

Competing Interests

Author Contributions

Data Availability

The authors declare there are no competing interests.

Xiaoxia Hu conceived and designed the experiments, performed the experiments, analyzed the data, prepared figures and/or tables, authored or reviewed drafts of the article, and approved the final draft.

Peng Wang performed the experiments, analyzed the data, prepared figures and/or tables, authored or reviewed drafts of the article, and approved the final draft.

Dali Zeng performed the experiments, authored or reviewed drafts of the article, and approved the final draft.

Guo-xin Hu conceived and designed the experiments, authored or reviewed drafts of the article, and approved the final draft.

The following information was supplied regarding data availability:

The raw measurements are available in the Supplementary File.

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
