# Peer review of "The effect of gene polymorphism on ticagrelor metabolism: an in vitro study of 22 CYP3A4 variants in Chinese Han population"

_PeerJ, doi:10.7717/peerj.18109_

## Round 0.1 · original submission · Minor Revisions

Overall, this was a straightforward study on CYP3A4 variants. Minor revisions are requested for the manuscript. In addition to the 3 reviewers' comments, please address the following:

The methods describe measuring amount of product produced over time, followed by calculating Km and Vmax (standard Michaelis–Menten kinetics). However, the results also provide inhibition constants (Ki). I am not sure how inhibition is calculated given the experimental approaches being used (there is no substrate-inhibitor competition). Most Ki values were in the high micromolar range that are far beyond the data presented in Figure 2 and are not physiologically relevant. Ki values were not mentioned in the methods or results sections, either. Therefore, please remove all Ki values, which do not add any meaning to the study and detract from the more relevant results (Km, Vmax, CL, etc.).

Discussion: At the end please mention how this data can extrapolate to in vivo situation and any limitations. Also mention the clinical concentrations of metabolite in plasma and how they compare to the in vitro data from this study.

Introduction: Please identify the biotransformation of parent to metabolite (e.g., hydroxylation, desmethylation, etc.)

Abstract and Figure 2: Please define CYP3A4*1 as the wild-type.

Tables 1 and 2: Please remove first column ("analytes") since it's uneccessary.

Reviewer 1 ·

Basic reporting

1. A work that potentially led to improvement in science. However, it is limited to just one community. If it can be expanded to a wider group, then this study will be exceptional. Probably can expand it to international collaboration in the future.

2. Work provides some idea for treatment strategies and risk assessments associated with ticagrelor much more downstream work needed in future.

3. Although the language is readable, it is advisable, to have native English speaker or engage a profesional English language service to improve it until it reaches profesional standards. Please check thoroughly on the typo, grammar and punctuation throughout the manuscripts. There are many careless mistake on it.

Experimental design

1. is the 22 CYP3A4 alleles is a part of 54 alleles reported and named by the Human Cytochrome
P450 Allele Nomenclature Committee? What is the reason justification for choosing only the 22 CYP3A4 alleles for these studies?

2. When the author referring to Chinese, which Chinese community are referred to? Is it just Chinese from China only? or to be more specific is it Han Chinese only? As even China, it's having diverse group of ethnicities. Author should make it clear in the manuscripts which chinese group they are referring to.

Or if the author referring to China population, the title manuscript should be change to China population from chinese.

Validity of the findings

No comments.

Additional comments

No Comment

Reviewer 2 ·

Basic reporting

1. The article suggests that adverse reactions are primarily related to ticagrelor concentration. However, AR-C124910XX is also a major active metabolite, with activity similar to ticagrelor. Therefore, considering only ticagrelor concentration when assessing adverse reactions may be biased. Please elaborate on the possible reasons in the introduction or discussion for why adverse reactions are related solely to ticagrelor and not to the concentration of its active metabolite AR-C124910XX.
2. Given that the common adverse reactions are primarily related to the blood concentration of ticagrelor, it is recommended that you specify the proportion of ticagrelor metabolized by the liver in the main text. This is important because multiple factors affect drug concentration, and if the proportion metabolized is low, this pathway might not be considered significant.

Experimental design

3. Please include the rationale for selecting these 22 variants in the manuscript. If the study focuses on the Chinese population, please provide the gene frequency of the 22 mutations in this population in the supplementary materials.
4. Considering that the construction of the bioanalytical method is crucial for this experiment, could you provide more details on the ARC124910xx UHPLC-MS/MS analysis method, including the standard curve settings and its range?
5. Why did you choose to set up three parallel samples? Could having fewer study groups affect the results?
6. In lines 191-195, while classifying CYP3A4 variants, the 50%, 80%, and 90% reduction in CLint were used as boundaries. Were these thresholds based on literature references or were they self-defined? Please cite the relevant references.

Validity of the findings

7. In the discussion, please summarize which of these variants have been previously studied for their polymorphic effects on ticagrelor metabolism. Additionally, indicate whether the findings of those studies align with the results presented in this paper.

Reviewer 3 ·

Basic reporting

The authors describe their work on the effect of gene polymorphism on ticagrelor metabolism: an in vitro study of 22 CYP3A4 variants in Chinese. This article first describes clinical adverse reactions to ticagrelor, such as bleeding and dyspnea, are related to the concentration of ticagrelor rather than its metabolites. The authors concluded that more attention needs to be given to the effect of gene polymorphism on ticagrelor concentration. This is an interesting and worthwhile study. Appropriate methodology has been employed and the conclusions appear to be justified based on the data at hand. It is recommended for publication in Peer J. I have a few recommendations for consideration.

Experimental design

1. Introduction. Please provide a clear hypothesis to be tested as well as a stronger rationale for the study.
2. Methods. Please indicate the purity level of the ticagrelor, AR-C124910XX and carbamazepine used in the study.
3. Methods. The reason chooses carbamazepine as an internal standard?
4. Discussion. Clarify the significance of the findings: Discuss the clinical implications of the observed variations in intrinsic clearance (CLint) among the different CYP3A4 genotypes. Explain how these variations may impact the pharmacokinetics of ticagrelor in the Chinese Han population.

Validity of the findings

1. Table. Do not use abbreviations in table headings. Write them out and give the abbreviation in parentheses or provide a footnote (e.g. RSD and RE in table 1&2)
2. General. The manuscript needs to be reviewed very carefully for English language/grammatical errors.

---

## Round 0.2 · Minor Revisions

One reviewer recommends accepting your article after addressing one additional comment. Please address the final comment below from the reviewer:

The authors mentioned that the thresholds of 50%, 80%, and 90% reduction in CLint for classifying CYP3A4 variants were self-defined based on the range of relative clearance values. Could the authors clarify the rationale behind selecting these specific boundaries? Were these thresholds scientifically justified?

Reviewer 1 ·

Basic reporting

No further comments. All Issue raised previously had been addressed accordingly by the authors.

Experimental design

Ammendment done accordingly

Validity of the findings

No further comment

Reviewer 2 ·

Basic reporting

The authors have carefully addressed the comments and thoroughly analyzed the content of the manuscript. I appreciate the clear presentation of the changes, which, in my opinion, have substantially improved the quality of the manuscript.

Experimental design

However, I do have a minor concern regarding the response. The authors mentioned that the thresholds of 50%, 80%, and 90% reduction in CLint for classifying CYP3A4 variants were self-defined based on the range of relative clearance values. Could the authors clarify the rationale behind selecting these specific boundaries? Were these thresholds scientifically justified?

Validity of the findings

No comment.

Additional comments

No comment.

---

## Round 0.3 · accepted · Accept

The authors adequately addressed the last minor comment